



**Soil nitrogen transformation responses to seasonal precipitation**
**changes are regulated by changes in functional microbial abundance**
**in a subtropical forest**
**Jie Chen[1,2,#], Guoliang Xiao[1,2,#], Yakov Kuzyakov[3,4], Darrel Jenerette[5], Ying**
**Ma[1,2], Wei Liu[1], Zhengfeng Wang[1], Weijun Shen[1,*]**
[1]Center for Ecological and Environmental Sciences, South China Botanical Garden,
Chinese Academy of Sciences, 723 Xingke Road, Tianhe District, Guangzhou
510650, PR China
[2]University of Chinese Academy of Sciences, 19A Yuquan Road, Shijingshan District,
Beijing 100049, PR China
[3]Department of Soil Science of Temperate Ecosystems, Georg August University
Göttingen, Büsgenweg 2, 37077 Göttingen, Germany
[4]Department of Agricultural Soil Science, Georg August University Göttingen,
Büsgenweg 2, 37077 Göttingen, Germany
[5]Department of Botany and Plant Sciences, Center for Conservation Biology,
University of California Riverside, Riverside, CA92521, USA
[#] These authors contributed equally to this work.
*Correspondence to*: Weijun Shen (shenweij@scbg.ac.cn)



**Abstract**
More dry-season droughts and wet-season storms have been predicted in subtropical
areas. Since subtropical forest soils are significant sources of $N_2O$ and $NO_3^-$, it is
important to understand the features and determinants of N transformation responses
to the predicted precipitation changes. A precipitation manipulation field experiment
was conducted to reduce dry-season precipitation and increase wet-season
precipitation, while keeping the annual precipitation unchanged in a subtropical
forest. Net N mineralization, net nitrification, $N_2O$ emission, nitrifying (bacterial and
archaeal *amoA*) and denitrifying (*nirK*, *nirS* and *nosZ*) genes abundance, microbial
biomass carbon (MBC) and soil physicochemical properties were monitored to
characterize and explain soil N transformation responses. Dry-season precipitation
reduction decreased net nitrification and N mineralization rates by 13 - 20%, while
wet-season precipitation addition increased both rates by 50%. More than 20% of the
total variation of net nitrification and N mineralization could be explained by
microbial abundance and soil water content (SWC), but archaeal *amoA* abundance
was the main factor. Increased net nitrification in wet season together with large
precipitation events caused substantial $NO_3^-$ losses via leaching. However, $N_2O$
emission decreased moderately either in dry or wet seasons due to changes in *nosZ*
gene abundance, MBC, net nitrification and SWC (decreased by 10 - 21%). We
conclude that reducing dry-season precipitation and increasing wet-season
precipitation affect N transformation mainly through altering functional microbial
abundance and MBC, which are further determined by changes in DOC and $NH_4^+$



availabilities. Such contrasting precipitation pattern will increase droughts and $NO_3^-$
leaching in subtropical forests.
**Key-words:** Denitrification, functional genes, nitrification, nitrogen cycle,
precipitation change, $N_2O$ emission



## 1 Introduction

Precipitation changes caused by global climate change are increasingly severe over the century (IPCC, 2007; Seager et al., 2007). Future projected precipitation patterns vary spatially and temporally, and the complexity and unpredictability of precipitation changes have exceeded other climate changes such as elevated $CO_2$ and temperature (Beier et al., 2012). Despite the frequency and intensity of precipitation events, seasonal precipitation changes are of increasing severity (Easterling et al., 2000). Recent study of 60 years precipitation data showed remarkable seasonal precipitation redistribution in a subtropical forest, with more frequent droughts in dry season and extremely rainfall events in wet season (Zhou et al., 2011). In contrast to annual precipitation amount, seasonal distribution may be more important in controlling the ecosystem functioning in subtropical forests, because of strong contrast between dry and wet seasons (Wang et al., 2009). Recent meta-analyses on precipitation manipulation experiments pointed out the lack of data in the warm and humid monsoon zones (Wu et al., 2011; Liu et al., 2016), and that more than 60% of all manipulative field experiments only focused on changes in precipitation amounts ((Beier *et al*., 2012). The consequences of seasonal precipitation redistribution at ecosystem levels are still under investigation. Altogether, field experiments simulating seasonal precipitation changes in subtropical regions are urgently needed for better understanding of the ecosystem responses.

Since hydrological cycle is highly coupled with soil biogeochemical cycles, changes in precipitation can strongly affect the nutrient transformations, particularly nitrogen (N) cycling and balance, thus exerting a feedback on climate (Davidson et al., 2008; Wieder et al., 2011). For instance, Annual $N_2O$ emission decreased by a rainfall exclusion in moist



tropical forests, but recovered within the first year after the rainfall exclusion was stopped
(Davidson *et al*. (2008). Net N mineralization rate declined sharply in response to increased
rainfall, but increased during drought in grasslands (Jamieson et al., 1998). Opposite
response patterns were also obtained in temperate forests (Emmett et al., 2004; Chen et al.,
2011; Fuchslueger et al., 2014). Nevertheless, limited information was known about the
responses of N cycle to seasonal precipitation changes in subtropical forests which serve as
important sources of $N_2O$ emission and inorganic N leaching (Fang et al., 2009; Isobe et al.,

2012).

Seasonal precipitation changes may disturb the natural seasonal dynamics of microbial
activities, soil moisture, temperature, plant nutrient uptake, carbon (C) and N availabilities,
and consequently the N transformations (Reichmann et al., 2013). Although the direct effects
of soil physicochemical properties and microbial communities on N transformations are well
documented, the predominant factors in determining N transformations under precipitation
changes are still debatable (Petersen et al., 2012; Auyeung et al., 2015).
Ammonium oxidation, the central and rate-limiting step in N cycle  is driven by
ammonia-oxidizing archaea (*AOA*) and bacteria (*AOB*), which are marked by the *amoA*
functional gene (van der Heijden et al., 2008). The release and consumption of $N_2O$ by
denitrification are mainly driven by nitrite-reducing bacteria marked by the *nirK* and *nirS*
genes and nitrous oxide-reducing bacteria marked by *nosZ* gene (Schimel and Bennett,
2004; Levy-Booth et al., 2014), respectively. Thus, changes in these microbial functions
can shed lights on the underlying mechanisms driving N transformation responses. The
abundance, composition and activity of these microbial functional groups largely depend



on soil moisture, temperature, $O_2$ diffusion, C and N availabilities - all of these factors are
strongly influenced by precipitation (Bell et al., 2014). For instance, reduced precipitation
decreases soil moisture and increases aeration and $O_2$ diffusion, which stimulates the
activity of nitrifiers (*AOA/AOB*) and nitrification, but constrain the activity of denitrifiers,
and consequently the $N_2O/N_2$ emissions (Stark and Firestone, 1995; Zhalnina et al., 2012).
However, both the denitrifiers and nitrifiers can be suppressed by decreased moisture and
available C during drought (Bárta et al., 2010; Zhalnina et al., 2012). In addition, increased
precipitation raises the $NH_4^+:NO_3^-$ ratio as $NO_3^-$ is easily leached (Reichmann et al., 2013),
and consequently alter the predominant microbial groups (Nautiyal and Dion, 2008). The
potential for mixotrophic growth and low substrate tolerance of nitrifying communities
(Levy-Booth et al., 2014) suggests a broader ecological niche occupied by the nitrifying
groups. Therefore, the nitrifying and denitrifying microorganisms may respond differently
to seasonal precipitation changes, leading to non-synchronously changes in nitrification
and denitrification, and consequently different changes in soil $NO_3^-$, $NH_4^+$ and $N_2O$ pools.
Nonetheless, the extent to which microorganisms control N transformations remains
unclear because soil physicochemical properties can also affect N pools through erosion,
leaching, plant uptake and physiological changes in microbial activity, regardless of
microbial composition or abundance (Cregger et al., 2014; Auyeung et al., 2015). As a
result, the effects of soil physicochemical properties and microbial communities on N
transformation rates are difficult to differentiate, which make it difficult to uncover the
underlying drivers.

In order to investigate responses of N transformations to seasonal precipitation changes



and the main controlling factors,  a precipitation manipulation experiment was conducted
in a subtropical forest in southern China, where the precipitation is projected to increase in
wet seasons and decrease in dry seasons (Zhou et al., 2011). We simulated similar seasonal
precipitation redistribution by reducing precipitation in dry seasons and increasing
frequency of large precipitation events in wet seasons over two years. Changes in soil
physiochemical properties, net N transformation rates, nitrifying (bacterial and archaeal
*amoA*) and denitrifying (*nirK, nirS* and *nosZ*) genes abundance were analyzed and
implicated in a hypothetical path model,  aiming to test the effects of soil physicochemical
properties and microbial abundance on N transformation rates (Fig. 1). The path
coefficients and model fitness were analyzed by structure equation model (SEM). We
hypothesized that (1) decreasing precipitation in the dry season will reduce N
transformation rates via decreasing SWC, C and N availabilities and microbial abundance,
but (2) precipitation addition during wet season will have little impact on N transformation
due to the originally sufficient SWC and substrate supply; (3) The responses of N
transformation rates to precipitation change will be more influenced by functional
microorganisms than by other biotic and abiotic variables; (4) microbial abundance is
directly influenced by soil physicochemical properties, but denitrifiers will be more easily
affected than nitrifiers, because the nitrifiers has the potential for mixotrophic growth and
low N and C substrate tolerance.



## 2 Material and methods

### 2.1 Site description

The study site is located at the Heshan National Field Research Station of Forest
Ecosystem, Chinese Academy of Sciences (112°54′E, 22°41′N), Heshan City,
Guangdong province, southern China. This area has a pronounced wet season (April
to September) receiving 80% of the annual rainfall, and a dry season (October to
March) with only 20% of the annual rainfall. The soil is typical laterite (or Oxisols
based on the USDA soil taxonomy), developed from sandstone, and is easily leached.
This study was conducted in a 35-year old evergreen broadleaved mixed species
(*EBMS*) forest dominated by *Schima superba* and *Michelia macclurei*. The forest
consists about 30 woody species, with average tree height of 8 m, average diameter at
breast eight (DBH) of 9.5 cm, stem density of 1430 trees ha$^{-1}$, and basal area of 11.6
m$^2$ ha$^{-1}$.

### 2.2 Experimental design

A replicated manipulative experiment of precipitation reduction in dry season and
precipitation addition in wet season was employed for two years from October 2012
to September 2014. Eight 12 m × 12 m experimental plots were randomly assigned to
4 replicates of each of the 2 treatment types: the seasonal precipitation change
manipulation (hereafter precip-change) and the trenched control (hereafter control).
Distance between the adjacent plots was at least 2 m. Around the perimeter of each of
the 8 plots, a 60-80 cm deep trench was excavated and 1 m height PVC segregation



board was imbedded to reduce the potential for lateral movement of soil water from
the surrounding areas into the plots. The precipitation reduction and addition was
realized by throughfall exclusion and water addition facilities, respectively.
Throughfall exclusion and water addition facilities were established in the 4 precip-
change plots, but not in the control. The facilities included supporting structures,
rainout shelters and water addition subsystems (Fig. S1). Within each of the 4 precip-
change plots, 16 galvanized steel pipes (2.5-3 m length × 10 cm diameter) were
vertically fixed in concrete bases which were imbedded in soil for 60 cm depth, and
were welded together with 8 horizontal stainless steel frames (12 m length) at the top.
Rainout sheets were fixed in two stainless steel frames and hanged on the supporting
system with steel hook rivets. There were about 8-12 rainout sheets (with the width of
50-100 cm) within each precip-change plot depending on the density of tree stems.
The rainout sheets were made from polyethylene plastic with > 90% light
transmission and installed at approximately 1.5 m height above the soil surface. The
total area of all the rainout sheets was 67% of the plot area (i.e., 144 m$^2$). The sheets
were opened to exclude throughfall during dry season (October 1$^{st}$ to March 31) but
folded without throughfall exclusion during wet season (April 1$^{st}$ to September 30$^{th}$).
Therefore, we reduced about 67% of the full incoming throughfall in the dry season.
The intercepted rainfall was routed into an iron gutter placed at the lower slope of the
plots, and then drained outside the plot with PVC pipes.

The water added into precip-change plots in the wet season was pumped from a

pond (about 800 m away from the experimental plots) and transported with PVC




pipes to the rubber sacs fixed on the supporting system, and then sprinkled out via 25
sprinklers distributed evenly in each plot. The pH was similar in the throughfall (6.42)
and pond water (6.19), and no differences of the nutrient (e.g. nitrogen and organic
carbon) contents between the pond water and throughfall were detected. The amount
of water added into a precip-change plot during the wet season was calculated as a
product of the above-canopy dry-season rainfall, the throughfall ratio, and the
throughfall exclusion ratio (i.e. 0.67). The above-canopy rainfall was obtained from a
standard meteorological station (Davis, Vaisala, Finland) about 80 m away from the
experimental site. The throughfall ratio was 0.86 obtained from 8 rain gauges
(TB4MM, Techno Solutions, Beijing, China) installed about 80 cm above soil surface
in the 8 plots. As a result, the intensity of the dry season rainfall events was reduced
and the frequency of large rainfall events in wet season was increased, while the
annually total quantity of the throughfall was not changed. More specifically, the
throughfall excluded was 220 mm in the 2013 dry season (Oct $1^{st}$ 2012 to Mar $31^{st}$
2013) and the same amount water was added back into each PC plot with 4 large
events (55 mm day$^{-1}$) in June through September 2013 (i.e., each event in one month)
to mimic the projected occurrence of more large rainfall events in wet season in the
region (Zhou et al., 2011). The throughfall exclusion was 170 mm in the 2014 dry
season (Oct $1^{st}$ 2013 to March $31^{st}$ 2014) and the same amount water was added back
into each precip-change plot with 3 large events (57 mm day$^{-1}$) in June through
August 2014 (Fig. 2).



### 2.3 Soil sampling and analysis


Soil samples were collected at the beginning and end of January, March, May, August
and October from May 2012 to September 2014 for physicochemical properties, and
from January 2013 to September 2014 for microbial functional genes analyses. Soil
samples were collected from 0 to 10 cm depth with an auger (Φ35 mm), sieved
through a 2 mm mesh to remove litter and stones. One composite soil sample,
consisting of six subsamples randomly collected within each plot, was used for the
physicochemical (stored at 4 °C) and microbial (stored at -20 °C) analyses. All
samples were analyzed within two weeks.
Soil physicochemical properties were measured using the methods as described by
Liu *et al*. (1996). Briefly, soil water content (SWC) was obtained by drying fresh soils
in an oven at 105 °C for 24 h. Total nitrogen (TN) and total phosphorus (TP) were
determined using the $H_2SO_4$ digestion-indophenol blue colorimetry and $H_2SO_4$
digestion-Mo-Sb colorimetry methods, respectively. $NH_4^+$ and $NO_3^-$ contents were
determined from the 2 M KCl extraction liquid by using the indophenol blue
colorimetry and copperized cadmium reduction methods, respectively.
Soil dissolved organic carbon (DOC) and microbial biomass carbon (MBC) were
measured immediately after the soil sampling using the fumigation extraction method
described as Vance, Brookes & Jenkinson (1987). In detail, a pair of fresh soil
subsamples (10 g) was placed into two glass breakers. One was fumigated in a
vacuum dryer with alcohol-free chloroform and NaOH solution for 24 h in dark, and
the other one was placed in dark for 24 h without fumigation. The two subsamples



were extracted with 0.5 M $K_2SO_4$ after fumigation, and the DOC concentration was
determined using a total organic C analysis instrument (TOC-VCSH, Shimadzu,
Japan). The difference of DOC concentration between the fumigated and un-
fumigated was multiplied by 0.45 to calculate MBC content.

Soil total DNA was extracted from 0.3 g fresh soil using the HiPure Soil DNA Mini

Kit (Magen, Guangzhou, China), quantified with a NanoDrop 2000
spectrophotometer (Thermo Fisher Scientific Inc., USA) and stored at -20 °C for
further analyses. The abundance of bacterial and archaeal ammonia-monooxygenase
gene (*amoA*), nitrite reductase genes (*nirK* and *nirS*) and nitrous oxide reductase gene
(*nosZ*) were quantified by using absolute Real-time PCR on an ABI 7500
thermocycler system with primers and thermal profiles presented in the
supplementary material (Table S1). The Real-time PCR reactions was performed on
96-well plates (Axygen, USA), with 20 ml volume in each well including 12.5 μl
SYBR Premix Ex Taq (TaKaRa Biotechnology, Japan), 1 μl of each primer (10 mmol
$L^{-1}$), 2 μl of DNA template (10 ng), 1 μl Dimethyl sulfoxide and 4.5 μl double-
distilled water. Standard curve was generated from a tenfold serial dilution ($10^3$-$10^8$
copies per μl) plasmid extracted from clones containing the target genes fragment for
the calculation of functional genes abundance in each sample.
**2.4    Measurement of N transformation rates**
Net N mineralization and nitrification rates were measured through the *in situ* field
soil incubation using the resin-core method (Reichmann et al., 2013). Six paired soil





cores (0-10 cm) were randomly sampled within each plot at the beginning of January,
March, May, August and October from May 2012 to September 2014. One core of
each pair was sieved through a 2-mm sieve after removing litter and stones, and
stored at 4 °C  for the initial pre-incubation measurements of SWC, $NO_3^-$ and $NH_4^+$.
The other core was incubated for one month in a PVC pipe that was open on both
sides and was oriented vertically with an ion exchange resin bag placed at the bottom
to collect inorganic N leached from the core. Soil cores and resin bags in the PVC
pipes were collected after the one-month incubation, and the soil was sieved and
stored at 4 °C  for the final post-incubation measurements of SWC, $NO_3^-$ and $NH_4^+$.
The net N mineralization rate was calculated as the final $NO_3^-$ and $NH_4^+$ content
minus the initial $NO_3^-$ and $NH_4^+$ content, and the net nitrification rate was calculated
as the final $NO_3^-$ content minus the initial $NO_3^-$ content (Reichmann et al., 2013).
Concentrations of $NO_3^-$ and $NH_4^+$ extracted from the resin were considered as the
leaching rates of $NO_3^-$ and $NH_4^+$ per month.
Nitrous oxide ($N_2O$) fluxes from soils were measured twice per month, from
October 2012 to September 2014, using static chamber and gas chromatography
techniques. The static chambers were made from white PVC materials and consisted
of a removable cover box and a base. The removable cover box with diameter of 26
cm and height of 35 cm, was an open-bottom PVC pipe, equipped with a 12 V fan on
the internal top wall to make turbulence sufficiently during gas sampling. The base of
the static chamber was nested together by an inside (25 cm diameter × 11 cm height)
and an outside (33 cm diameter × 8 cm height) PVC pipes, with a water groove left



between the two pipes for sealing during gas samples collection. The bottom of the
base was cut sharply to facilitate soil insertion. Two months before gas sampling, four
static chambers were deployed randomly at each plot to minimize effects of
installation disturbance.
The $N_2O$ samples were collected between 09:00 and 11:00 a.m. local time. Prior to
gas sampling, the cover box was placed on the collar filled with water in the groove,
and the fan was turned on simultaneously. The static chamber was closed for 30
minutes, and gas samples were taken using 100 ml plastic syringes at the initial closed
time as well as every 10 minute thereafter during the closed period. When collecting
gas samples, the soft rubber hose connected with static chamber was cleaned
thoroughly by pumping plastic syringe for three times, then 80 ml gas sample inside
the chamber was collected and transferred into a 500 ml polyethylene-aluminum
coated gas sampling bag. At the same time, values of atmospheric pressures and air
temperatures inside static chambers were measured for three times. After gas
sampling, cover boxes were removed to reduce disturbance to experimental plots as
much as possible. $N_2O$ concentrations were analyzed in the laboratory by gas
chromatography (Agilent 7890A, Agilent Technologies, USA) equipped with an
electron capture detector set at 300 °C and a stainless porapak-Q column set at 70 °C
within 24 hours following gas sampling. $N_2$ was used as carrier gas at the flow rate of
30 ml min$^{-1}$. The $N_2O$ concentration of standard gas for system calibration was 332
ppbV. The $N_2O$ flux was calculated by changes of $N_2O$ concentrations inside static
chamber during periods of gas sampling, with the equation as follows:



$$\mathbf{F} = \rho \times \frac{V}{A} \times \frac{P}{P0} \times \frac{T0}{T} \times \frac{dC}{d_t}$$
Where F stands for the flux of $N_2O$ (mg m$^{-2}$ hr$^{-1}$), ρ stands for the density of $N_2O$
under standard condition (g L$^{-1}$), V stands for the effective volume of chamber (m$^3$), A
stands for the area of soil covered by chamber (m$^2$), P and T stand for the atmospheric
pressures (Pa) and absolute air temperature inside chamber (K) when gas sampling, $P_0$
and $T_0$ stand for the atmospheric pressures (Pa) and the absolute temperature (K)
under standard condition, and $\frac{dC}{dt}$ stands for changes of $N_2O$ concentrations in the
chamber during gas sampling.
## 2.5 Statistical analysis
Two-way repeated-measures analysis of variance (ANOVA) with sampling time as the
repeated factor was used to examine the effects of precip-change and sampling time
on all measured parameters. Pillai's trace from multivariate test was used for within-
subjects test when the assumption of multisample sphericity was not met. Independent
samples *t* tests were used to detect the difference of each variable between precip-
change and control at each sampling time. All the parameters were explored for
normality (Kolmogorov-Smirnov test) and homogeneity of variances (Levène test)
prior to the analyses, and log-transformed If necessary. All statistical analyses
described above were performed using SPSS v.16.0 (SPSS Inc., Chicago, IL, USA).
Structural Equation Modeling (SEM) were performed with AMOS 21.0 (SPSS Inc.,
Chicago, IL, USA) to test the hypothetical causal relationships among soil
physicochemical properties, microbial abundance and N transformation rates in the



conceptual model (Fig. 1). How the effects of soil physicochemical properties and
microbial abundance determine the responses of N transformation rates were
evaluated. In order to explicitly illustrate the pathways of soil physicochemical
properties and microbial abundance involved in each N transformation process, three
individual models were constructed corresponding to the conceptual model to explain
the responses of (a) net nitrification, (b) net N mineralization and (c) $N_2O$ emission
rates. Three models may be easier to discover the controlling factors than one
complex model which implicates all the measured processes. In these models, the
precip-change treatments are categorical exogenous variables with two levels: 0
representing control and 1 representing seasonal precipitation changes (Delgado-
Baquerizo et al., 2014). Abundance of both *nirK* and *nirS* genes were evidenced
correlated with nitrification or N mineralization rates (Levy-Booth et al., 2014).
Therefore, *nirK* and *nirS* abundance were added as one (*nirK+nirS*) endogenous
factors in model. Net nitrification rate was included in model (b) as an endogenous
factor because it may influence $N_2O$ emission through altering the production of $NO_3^-$
as the substrate for $N_2O$ production. Prior to the SEM analyses, normal distribution of
all the involved variables were examined, and genes abundance were log-transformed.
Goodness of model fits was evaluated by chi-square test ($p > 0.05$), comparative fit
index (CFI > 0.95), and root square mean errors of approximation (RMSEA < 0.05)
(Hu and Bentler, 1998; Schermelleh-Engel et al., 2003). Pathways without significant
effects were not shown ($p > 0.05$) in the final models.

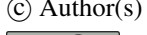



## 3 Results

### 3.1 Responses of soil physicochemical properties, N transformation rates and microbial abundance to precipitation changes

Before the precipitation manipulation from May to September in 2012, average net N transformation (i.e. N nitrification, mineralization and leaching) rates, N ($NO_3^-$, $NH_4^+$, TN) and organic C (MBC, DOC, TOC) contents as well as soil temperature were similar among all plots (Table S2). In the two dry seasons with precipitation reduction, SWC decreased by 16 % in 2013 and by 21 % in 2014 ($p < 0.01$, Fig. 2, Table S3). Similarly, $NO_3^-$ concentration decreased by 35 % and 24 % in 2013 and 2014, respectively ($p < 0.01$, Fig. 2, Table S3). Opposite patterns were observed for $NH_4^+$ concentration, which increased with the precipitation reduction (Fig. 2). In the wet seasons with precipitation addition, SWC, $NO_3^-$ concentration, DOC and MBC remained lower in the precip-change plots than in the control plots in both years (Fig. 2, Table S3).

Precipitation reduction strongly decreased the average dry-season net nitrification rate by 13 % in 2013 and by 20 % in 2014, and decreased net N mineralization rate by 16 % in 2013 and by 18 % in 2014 ($p < 0.1$, Fig. 3, Table S4). The $NO_3^-$ leaching also declined with precipitation reduction, especially in 2014 with a marked decrease by 22 % ($p < 0.001$, Fig. 3, Table S4). Contrastingly, the rates of three N transformation processes increased by 50% with precipitation addition in the 2013 wet season whereas changed little in the 2014 wet season (Fig. 3). Throughout the two years, moderate decreases were detected in $N_2O$ emission either during dry-season precipitation





reduction (35%) or during wet-season precipitation addition (15%) (Fig. 3, Table S4).
No amplification of bacterial *amoA* gene was detected in soil neither from the
precip-change plots nor from the control plots, which was mainly because soil *AOB*
community abundance in the studied forest was under the detection limit (Isobe et al.,
2012). The average seasonal archaeal *amoA* gene was $6.5 \times 10^6 \pm 1.9 \times 10^6$ copies g$^{-1}$
dry soil, and varied significantly according seasonal precipitation changes. With
precipitation reduction, the archaeal *amoA* gene abundance changed little in the 2013
dry season but decreased by 70% in the 2014 dry season (Fig. 4). The abundance of
three denitrifying genes (*nirK*, *nirS* and *nosZ*) increased with precipitation reduction
by 30-80% in the 2013 dry season ($p < 0.05$, Fig. 4, Table S5). In both seasons of
2014, neither dry-season precipitation reduction nor wet-season precipitation addition
had significant impacts on the abundance of the three denitrifying genes (Fig. 4, Table
S5).

## 3.2    Paths determining N transformation rates and functional microbial abundance

Although the annual precipitation amount was kept constant, the redistribution of
seasonal precipitation imposed an overall negative impact on SWC and NO$_3^-$
concentration (Fig. 5). SWC affected net nitrification and N mineralization through a
direct negative path and N$_2$O emission through a direct positive path (Fig. 5). Net N
mineralization, nitrification and N$_2$O emission rates were also affected by the functional
genes abundance and MBC paths. Since bacterial *amoA* gene was not detected, we only



use the archaeal *amoA* abundance as the dominant nitrifying microbial abundance in
the SEM analyses. Specifically, the archaeal *amoA* gene abundance and MBC had direct
positive impacts on net N mineralization and nitrification rates, whereas the *nosZ* gene
abundance had a direct negative impact on $N_2O$ emission (Fig. 5). As a result, 21% and
22% of the net N mineralization and nitrification variability are explained, respectively
(see the $r^2$ in Fig. 5). Among the direct influential factors, archaeal *amoA* abundance
showed the strongest correlations either with net N mineralization or with net
nitrification rates. Soil $N_2O$ emission was mostly affected by positive effects of net
nitrification rate and SWC, followed by negative effects of *nosZ* abundance and MBC,
and as much as 42% of the total variation could be explained (see the $r^2$ in Fig. 5).
Precip-change-induced changes in SWC had no direct impacts on functional genes
abundance. Instead, the functional genes abundance was indirectly affected by the
precip-change-induced alterations in $NO_3^-$, $NH_4^+$ concentrations and DOC (Fig. 5).
Specifically, $NO_3^-$ and $NH_4^+$ had direct positive effects on archaeal *amoA* abundance
whereas DOC had a direct negative effect on *nirK+nirS* abundance. Both $NH_4^+$ and
DOC concentration had direct positive impacts on the *nosZ* abundance (Fig. 5).
Changes in MBC were directly positively influenced by SWC and DOC.
## 4   Discussion
### 4.1   Drivers of N transformation processes
Consistent with our hypotheses, seasonal precipitation redistribution induced
significant changes in net N mineralization and nitrification rates through altering SWC,



MBC and archaeal *amoA gene* abundance. $N_2O$ emission decreased moderately either
in precipitation reduction or addition, which indicated that soil N loss by $N_2O$ emission
in subtropical forests would be alleviated by the predicted seasonal precipitation
changes. In contrast, increased $NO_3^-$ leaching during precipitation addition in wet
season led to a significant loss of soil $NO_3^-$ pool. During the two years' experiment,
SWC was always lower in precip-change plots than in control plots, despite of the
precipitation addition in the wet seasons (Fig. 2). One reason is the higher transpiration
loss resulting from bigger trees in the precip-change plots than in the control plots. The
average tree height and DBH were respectively $10.2 \pm 5.0$ m and $10.7 \pm 6.3$ cm in the
four precip-change plots with the total number of 64 tree individuals, compared to 7.7
$\pm 3.5$ m and $9.5 \pm 5.2$ cm in the four the control plots having the total number of 68 tree
individuals.
Initially, we hypothesized that decreased precipitaiton in the dry season would
suppress N transformation, and precipitation addition during wet season would have
little impact on the N transformation processes because the soils are water saturated and
substrate sufficient. Agreeing with the first hypothesis, the net nitrification and N
mineralization rates decreased sharply with the reduction of throughfall in dry season
(Fig. 3). However, disagreeing with the second hypothesis, the nitrification and N
mineralization rates increased markedly during precipitation supplement in the wet
seasons (Fig. 3). These results were caused by the interactions among microbial
abundance, soil moisture and substrate availability (Fig. 5 and S3). Specifically, DOC
of the dry season was less available in the precip-change plots than in the control plots



(Fig. 2), probably due to reduced C input by less root production and exudation after
drying (Kuzyakov and Domanski, 2000; Borken and Matzner, 2009). The reduced soil
C substrate (or DOC) could suppress the growth of soil microorganisms (e.g. MBC and
*AOA*), and therefore resulted in decreased net nitrification and mineralization rates (Fig.
3 and 5). Although increased $NH_4^+$ concentration during precipitation reduction could
provide more N substrate for nitrifier, the negative effects of decreased SWC and DOC
may have overwhelmed the positive effects of increased $NH_4^+$ on microbial nitrification
process in dry season. Instead, the accumulated $NH_4^+$ after dry season precipitation
reduction might have a positive legacy effect on soil microbial activity in wet season,
leading to increased N transformations. In addition, SWC also directly affected N
transformations by physiological changes in microbial activity, regardless of microbial
abundance and composition (Auyeung et al., 2015). The increased N transformation
rates in respond to decreased SWC, MBC and archaeal *amoA* gene abundance during
precipitation addition might be one of such cases. A 10% decrease of SWC in the natural
humid wet season might create a better redox conditions for microbial nitrification, as
excessive soil moisture could reduce soil oxygen concentration. According to Borken
& Matzner (2009), the increases of soil microbial activity by rewetting usually occurred
with an increased pulse in reconstituting mineralization of SOM as well as an increase
of organic substrate availability. This study revealed substantial decrease in MBC and
archaeal *amoA* gene abundance, which indicate that microorganisms may reduce
microbial abundance and release the MBC and MBN from dead or non-active
microorganisms to support the increased energy demand caused by increased microbial

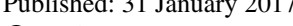

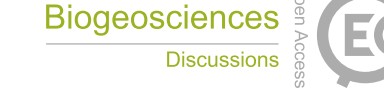

activity and accelerated microbial processes (Borken and Matzner, 2009).
We also hypothesized that the N transformation processes are predominantly
influenced by functional microbial abundance. As expected, net N mineralization and
nitrification rates showed stronger relationships with archaeal *amoA* abundance than
that with MBC and other soil properties (Fig. 5). However, MBC and denitrifying gene
abundance had similar effects on $N_2O$ emission, and only *nosZ* gene abundance exerted
a pronounced effect on $N_2O$ emission (Fig. 5), which may through reducing the $N_2O$
consumption (Henderson et al. 2010; Levy-Booth et al. 2014). No significant
correlation between $N_2O$ emission and *nirK + nirS* gene abundance was detected, which
was inconsistent with previous researches (Levy-Booth et al., 2014). The $N_2O$ emission
related denitrification can be also performed by nitrifiers and fungi in soils with high
aeration and limited substrate availability (Levy-Booth et al. 2014). The experimental
seasonal precipitation strongly decreased SWC and DOC content (Fig. 1), which could
lead to higher aeration while lower substrate availability, and consequently predominant
roles of nitrifier and fungi denitrification in controlling $N_2O$ emission. In addition, SWC
and nitrification rate also directly affected $N_2O$ emission via altering substrate
availability and consequent microbial activity, despite of high microbial abundance (Fig.
5). Overall, net nitrification and N mineralization were mainly regulated by *AOA*
abundance, while the controlling factors of $N_2O$ emission were complex.
**4.2   Determinants of nitrifying and denitrifying gene abundance**
The responses of both nitrifying and denitrifying genes were mainly related to the



changes in substrate concentrations. SEM analysis showed that both *amoA* and *nosZ*
genes abundance were positively affected by DOC and $NH_4^+$ concentration,
suggesting substrate constraints for these two functional microbial groups. This
disagreed with previous studies reporting that *AOA* community had the potential of
mixotrophic growth and low substrate tolerance when compared with its counterpart
*AOB* (Erguder et al., 2009; Shen et al., 2012). However, these results were mainly
caused by a stronger competitiveness of *AOA* than its counterpart *AOB*, as these
studies mainly focused on the relative effects of substrate availability on *AOA* and
*AOB*. Both *nosZ* and *amoA* genes abundance increased with DOC and $NH_4^+$
concentration (Fig. 5), which indicated *AOA* community could be constrained by C
and N substrates when competing with other microbes that have different functions.
Otherwise, the existing *AOA* species that have the potential ability of mixotrophic
growth and low substrate tolerance may not dominante in the studied subtropical
forest, as the soil is originally rich in SOM (Zhou et al., 2006; Chen et al., 2015).
Therefore, *AOA* community in the studied soil could be easily influenced by changes
in soil C and N availability.

The abundance of *nirK* and *nirS* genes was positively controlled by soil $NH_4^+$

concentration and negatively controlled by DOC content (Fig. 5). This further
confirmed that more $NH_4^+$ content could favor more abundant microorganisms
containing *nirK* or *nirS* genes (Yi et al., 2015), because higher $NH_4^+$ concentration
could supply sufficient $NO_3^-$ as the direct substrate or optimum pH value for growth
of the denitrifying microorganisms. The negative effect of DOC on *nirK* and *nirS*



gene abundance was inconsistent with most of previous reports that denitrifiers are
primary heterotrophic (Bárta et al., 2010). One reason is because high DOC
concentration can constrain the growth of microorganisms containing *nirK* and *nirS*
genes through effecting other factors, such as pH and C:N ratio (Henderson et al.,
2010; Levy-Booth et al., 2014). Generally, abundance of both nitrifying and
denitrifying gene abundance changed with precipitation redistribution, and the
direction and magnitude of the changes depended mostly on soil N and C substrate
availabilities.
**5   Conclusions**
To summarize, soil net nitrification and N mineralization rates responded significantly
to seasonal precipitation redistribution, and more than 20% of the variation could be
explained by the effects of microbial abundance, SWC, soil C and N substrates. *AOA*
community abundance was the main factor in regulating these two N transformation
processes. $N_2O$ emission during the two years' seasonal precipitation redistribution
decreased moderately, and as much as 42% of the total variation in $N_2O$ emission was
attributed to the total effects of SWC, nitrification rate, MBC and *nosZ* gene
abundance. The accumulated $NH_4^+$ due to precipitation reduction may stimulate
nitrification process in wet season, and consequently accelerate N loss from $NO_3^-$
leaching. Therefore, long term of the predicted seasonal precipitation changes in
subtropical forests may result in profound changes in different N pool size, with less
$N_2O$ emission while more $NO_3^-$ leaching, which in turn exert a feedback to climate
and environmental changes. Meanwhile, changes in functional microbial abundance



induced by soil DOC and $NH_4^+$ substrate availabilities will be the predominant driver
in regulating the extent and direction of soil N transformation changes.

## Author contribution

Jie Chen and Guoliang Xiao carried out the experiment, analyzed the data and wrote
the draft manuscript. Weijun Shen conceived the study. All authors contributed to
manuscript writing and revision.

## Acknowledgements

We thank Mr. Y. Lin, Z. Chen, M. Li and S. Fu for their helps on field works; Mrs. C.
Long and X. Zhou for their helps on laboratory assays. Financial supports came from
the National Natural Science Foundation of China (31130011, 31290222 and
31425005) and the Natural Science Foundation of Guangdong Province, China
(S2012020011084).

## Competing interests

The authors declare that they have no conflict of interest.





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



**Figure captions**
**Fig. 1.** A conceptual model illustrating the effects of physiochemical properties and
functional microorganisms on N transformation rates. Soil water content (SWC),
ammonium ($NH_4^+$), nitrate ($NO_3^-$) and dissolved organic carbon (DOC)
concentrations were included in the group of soil physiochemical property. Microbial
biomass carbon (MBC), nitrifying (*amoA*) and denitrifying (*nirK*, *nirS* and *nosZ*) gene
abundance were included in the microbial attributes group. The solid lines with
arrows indicate the direction of the effect.
**Fig. 2.** Seasonal dynamics of precipitation and soil physiochemical properties in
control and precip-change plots over the course of experiment. Points and bars with
standard error ($n = 4$) show mean values at each sampling time and in dry (DS) and
wet (WS) seasons. Grey shades indicate the periods of precipitation reduction. The
significance levels are presented as: $*p < 0.05$.
**Fig. 3.** Nitrogen transformation rates measured in control and precip-change plots
over the course of experiment. Points and bars with standard error ($n = 4$) show mean
values at each sampling time and in dry (DS) and wet (WS) seasons. Grey shades
indicate the periods of precipitation reduction. The significance levels are presented
as: $*p < 0.05$.
**Fig. 4.** Copy numbers of archaeal *amoA*, *nirK*, *nirS* and *nosZ* gene per gram dry soil
measured in control and precip-change plots over the course of experiment. Points
and bars with standard error ($n = 4$) show mean values at each sampling time and in





dry (DS) and wet (WS) seasons. Grey shades indicate the periods of precipitation
reduction. The significance levels are presented as: $*p < 0.05$.
**Fig. 5.** Path diagrams demonstrating the effects of soil physicochemical properties
and functional genes abundance on net nitrification, N mineralization and $N_2O$ efflux
rates in response to precipitation change during two years. Numbers adjacent to
arrows are path coefficients, which indicate the relationships between the two
variables on both sides of the arrows. Solid and dash lines represent positive and
negative paths, respectively. The $r^2$ above or below each response variable in the
model denotes the proportion of variance which could be explained. Size of the lines
indicate significant levels of path coefficients.





**Fig. 1**

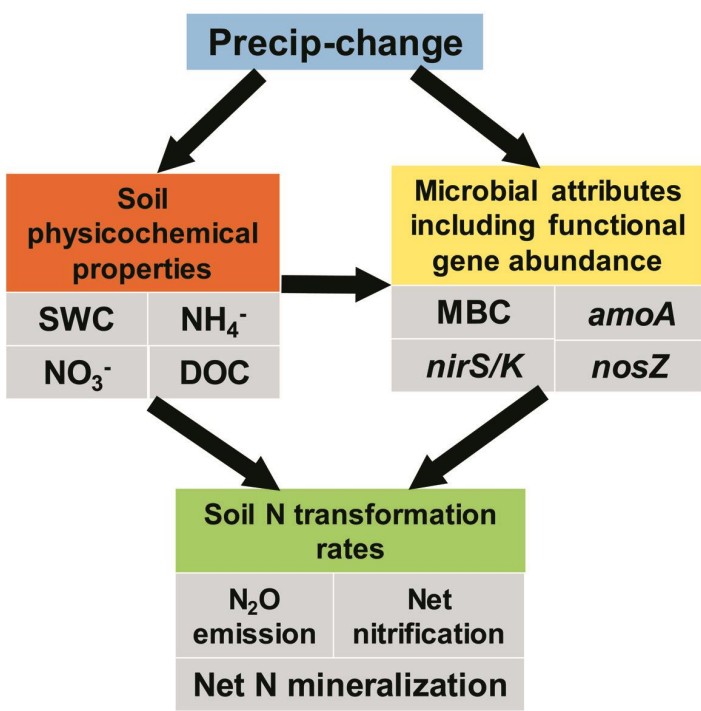




**Fig. 2**

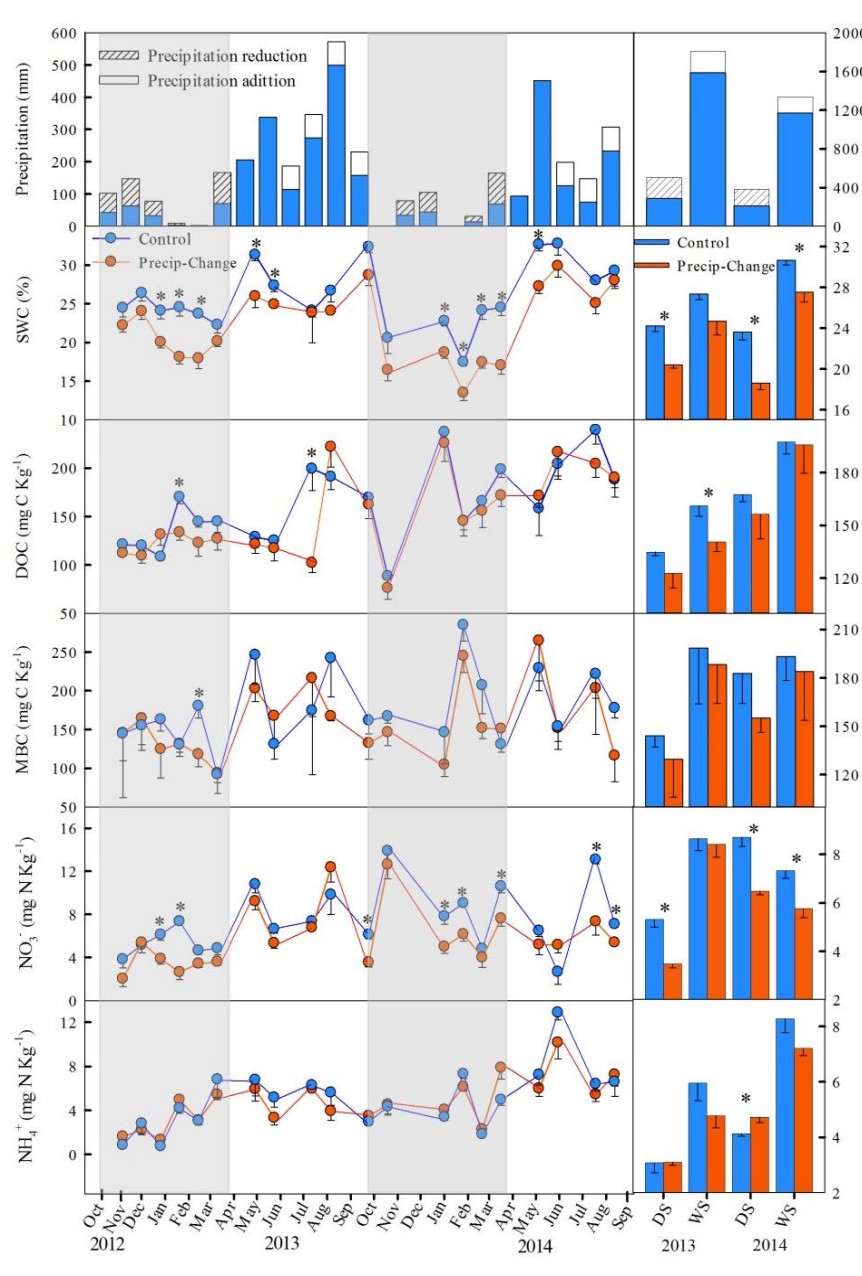




**Fig. 3**

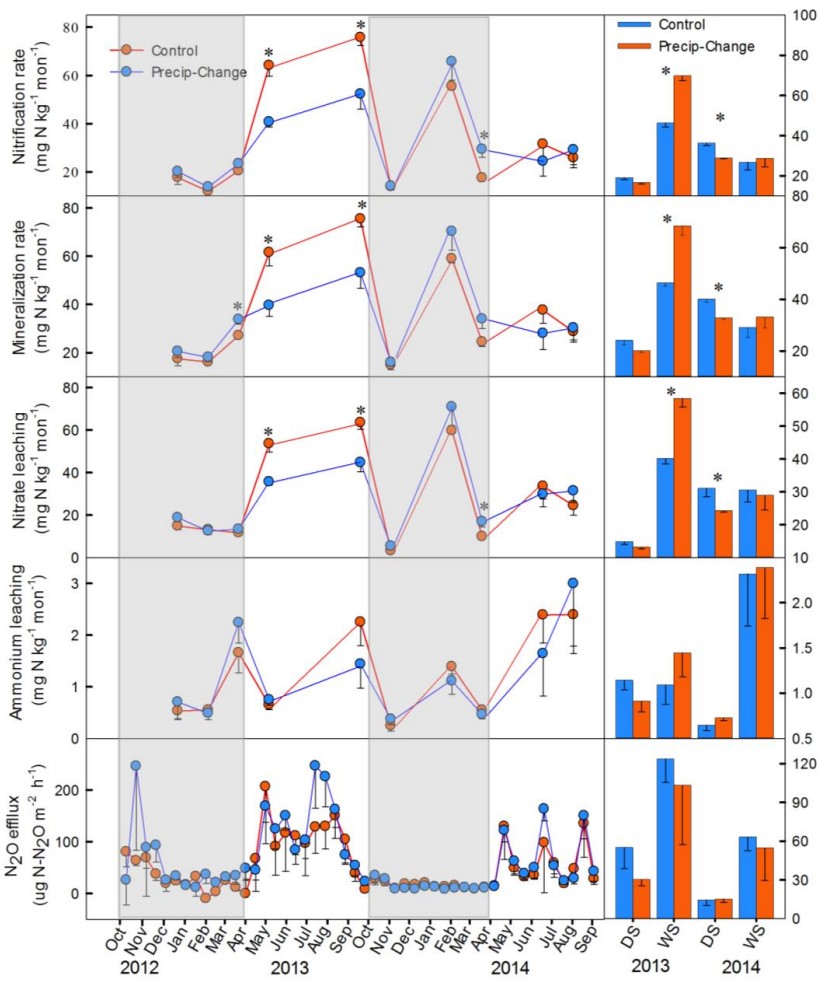






**Fig. 4**

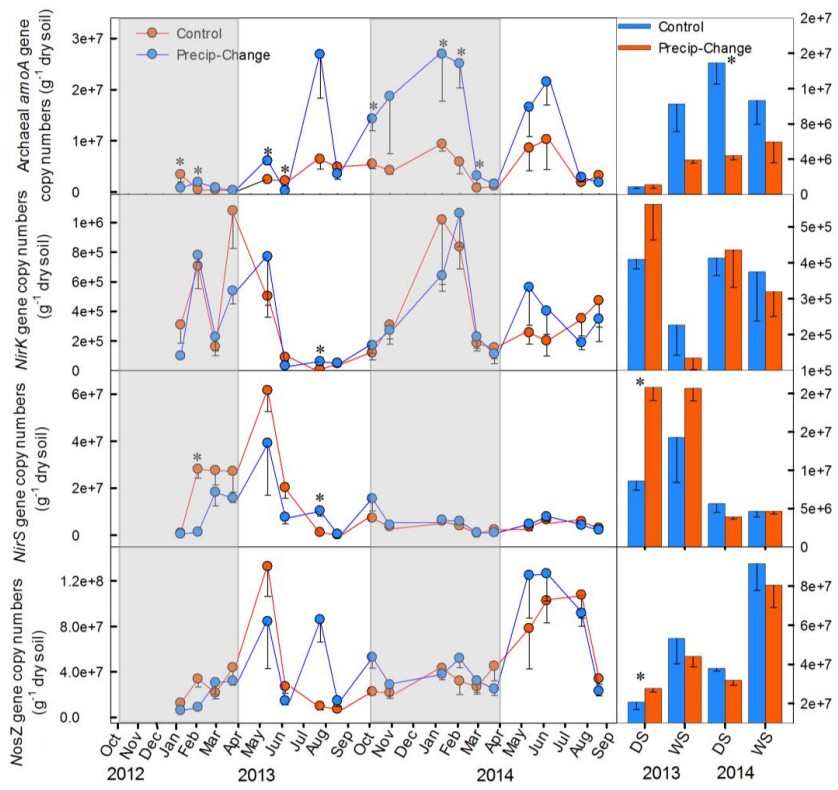






**Fig. 5**

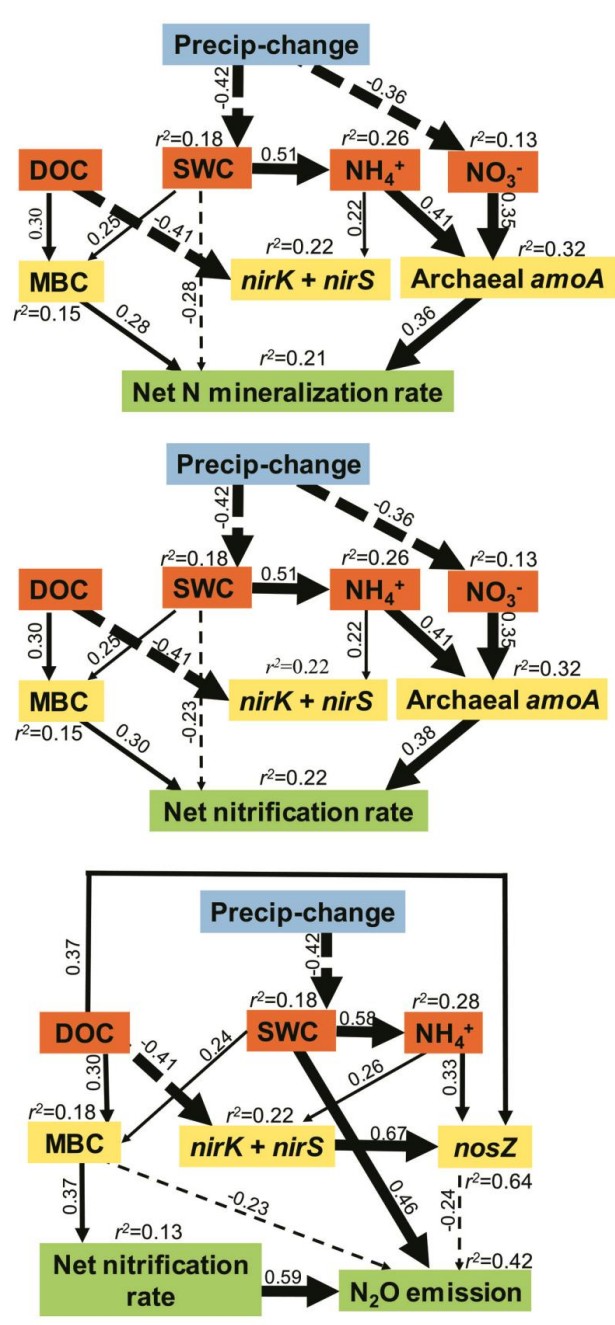