# Peer review of "Soil nitrogen transformation responses to seasonal precipitation"

_Biogeosciences, 2017_

## Referee Comment (RC1) · Anonymous Referee #1 · 4 Feb 2017

General: In this M.S, the authors took advantage of in-situ two-year rainfall manipulation experiment combining with monitoring of soil chemical, biological properties and nitrogen mineralization rates and N2O fluxes to study the features and determinants of nitrogen transformation. They found that more than 20% of the soil net nitrification and N mineralization rates variation could be explained by the effects of microbial abundance, SWC, soil C and N substrates. AOA abundance was the main factor in regulating these two N transformation processes, while as much as 42% of the total variation in N2O emission was attributed to the total effects of SWC, nitrification rate, MBC and nosZ gene abundance. The results suggested that predicted seasonal pre-

cipitation changes in subtropical forests might result in less N2O emission while more NO3- leaching. The study is of significant for efforts to understand the features and determinants of nitrogen transformation responses to the predicted precipitation change in subtropical area.

Specific: Line 44-45: it is not a proper conclusion Line 96: no verb of the sentence Line 123: reference? Need a brief introduction of SEM model. Lin 127-129: add the reason for the third hypothesis Line 137-139: add the data or reference Line 174-175: why not add rainwater? Line 183: in brackets, is that the instrument model of meteorological station? Line 255-292 too many sentences for the introduction of Nitrous oxide (N2O) fluxes measurement Line 297-298 :why not use paired sample T test? Line 395-402: Need further explanation why the precipitation addition treatment decreased soil water contents. How this SWC change will affect functional gene results. Fig. 1: precipitation change Fig. 2 mark the meaning of the blue bar; why the SWC is significantly lower under the precipitation addition treatment than the control in wet season? Fig. 4 why not use log transformed number? Fig. 5 add symbles a,b,c in the figure and M.S. Fig. S2 why analyze the relationships between nirK+nirS and archaeal amoA; between nosZ and nirk+nirS?

---

## Referee Comment (RC2) · Anonymous Referee #2 · 27 Feb 2017

This manuscript uses an interesting design to explore the complex relationships between changes in precipitation regimens and soil N dynamics. The authors monitored changes in microbial functional genes, N-mineralization rates, and soil N2O flux to understand how changes in precipitation regime, but not total precipitation amount, might affect soil biogeochemical cycles (i.e. N-cycle). Results indicated that 20% of total N-mineralization and nitrification rate were explained by microbial abundance (determined via fumigation) and soil moisture content, and that amoA abundance also helped to explain a significant portion of variation in measured N pools. Together, the results indicate that predicted changes in seasonal precipitation associated with

changing climate may result in increased NO3- leaching form soils and decreases in N2O emissions.

Although the manuscript is scientifically sound and interesting, it should be heavily edited for proper grammar, spelling, and sentence structure.

Specific Comments: Line 35-37: the wording of this sentence is confusing, please clarify "main factor" (i.e. is it a statistical inference that is meant here?) Line 39: do the authors mean "both dry and wet seasons"? this sentence should be clarified. Line 43: what was "determined by changes in DOC and NH4+"? please clarify this conclusion. Line 49: this sentence is missing a word, are the authors indicating that precipitation changes have been observed as increasingly sever, or that they are predicted to be increasingly severe. Line 54: how have changes in precipitation events have exceeded other climate change-related fluctuations? Line 57: seasonal redistribution of what metric? Line 123: reference for the SEM? Also, please introduce SEM and how/ if they have been previously used in a similar manner. Line 130: please clarify "easily" Line 135-145: please add a reference for these data, or indicate how the authors collected the data. Line 174: was this water also examined for microbial community structure and functional genes? Is it possible that this water acted as a source of microbial change? Line 231: why did the authors not also quantify bacterial and archaeal 16S rRNA genes? The assay would independently confirm results indicated by chloroform fumigation techniques, and may potentially help to shed light on the negative results associated with amoB gene quantification. Line 381: could the opposite also be true here? That alterations in DOC and N are due to changes in functional gene abundance? Line 398: reference? Line 424: is there a figure or table that would also demonstrate this statement? Line 433: please clarify this sentence, how are microorganism responsible for reducing microbial abundance? Line 444: are there other appropriate references available here? Figures 2-5 would benefit from the addition of panel indicators (i.e. "Fig1 A") to help calfiy to the reader exactly which data the authors are discussing in the MS. Fig 5: this is a nicely illustrated complex figure.

---

## Referee Comment (RC3) · Anonymous Referee #3 · 5 Mar 2017

The manuscript bg-2017-3 reported soil N transformation rates in relate to soil properties and microbial functional gene abundance in a rainfall-manipulation experiment in a subtropical forest. The authors showed that the rainfall manipulation (dry-season reduction and wet-season addition) increased $NO_3^-$ leaching and had minor effect on $N_2O$ emission, which can be linked to changes in soil properties and specific microbial functional genes (by SEM). This topic is very relevant to Biogeosciences, and the results are robust based on solid data (monitoring of soil properties, in situ N transformation rate measurements, and microbial functional gene measurements). I have few suggestions for the authors to consider in revision.

[Figure]

1. To be more accurate, DOC is EOC (extractable organic carbon).

2. The writing should be improved for clarity and readability. In some sections, it is wordy and not easy to follow the logic.

3. Microbial functional gene abundance has limited power in explaining the N transformation rates: a) many genes are involved in a process, and b) a gene is there does not mean it is expressed or "functioning". This needs to be mentioned.

4. L397-402: Why select these 8 plots for the experiment? The initial differences in stand characteristics between treatments should be minimized for such experiment.

5. Figure 1 and 5: why soil pH was not included in the SEM? Was it measured?

6. Figure 2: Why rainfall addition in the wet season did not lead to expected increase in soil water content, in both years? Also, MBC was not significantly affected by the rainfall treatment even in the dry season, which is unexpected. Do you have continuous measurement of soil moisture (by TDR or Decagon sensors) in these 8 plots?

7. Table S2: stand characteristics (species composition, stem density, tree height, basal area, etc.) should be included.

―――――――――――――――――――――――

---

## Author Comment (AC1) · 20 Mar 2017

General Comment:

In this M.S, the authors took advantage of in-situ two-year rainfall manipulation experiment combining with monitoring of soil chemical, biological properties and nitrogen mineralization rates and N2O fluxes to study the features and determinants of nitrogen transformation. They found that more than 20% of the soil net nitrification and N mineralization rates variation could be explained by the effects of microbial abundance, SWC, soil C and N substrates. AOA abundance was the main factor in regulating these

two N transformation processes, while as much as 42% of the total variation in N2O emission was attributed to the total effects of SWC, nitrification rate, MBC and nosZ gene abundance. The results suggested that predicted seasonal precipitation changes in subtropical forests might result in less N2O emission while more NO3- leaching. The study is of significant for efforts to understand the features and determinants of nitrogen transformation responses to the predicted precipitation change in subtropical area. Response: Thanks to the reviewer for the concise summary of our work. We acknowledged the reviewers for their constructive comments in the Acknowledgement section. The revised manuscript is attached as a Supplement to the point-by-point responses to reviewers comments. All the changes have been marked in the revised version..

Specific Comments:

Comment #1: Line 44-45: it is not a proper conclusion Response: We agree with the reviewer that the conclusion is too speculative and therefore has been deleted (see lines 47-48). The ending sentence of the abstract has also been revised accordingly.

Comment #2: Line 96: no verb of the sentence Response: The grammatical error has been fixed (see line 106).

Comment #3: Line123: reference? Need a brief introduction of SEM model. Response: Following the reviewer's suggestion, we revised the sentence and added two references. A brief introduction of the SEM model has been added in the statistical part (see lines 142, 332-337).

Comment #4: Line 127-129: add the reason for the third hypothesis Response: We revised the third hypothesis as "The responses of N transformation rates to precipitation change will be associated with the responses of functional microbial abundance" (see lines 146-147). The reason for proposing it has also been added (lines 148-149). According to Levy-Booth et al. (2014), soil N transformation processes are biologically catalyzed by specific enzymes coded by functional genes within functional microorganisms. For instance, ammonium oxidation is catalyzed by ammonium oxidase which

coded by the amoA gene in ammonia-oxidizing archaea (AOA) and bacteria (AOB) (van der Heijden et al., 2008). It has been well documented that N transformation rates are significantly related to functional microbial abundance represented by functional gene abundance in various environments (Petersen et al., 2012; Wertz et al., 2012). Levy-Booth, D.J., Prescott, C.E., and Grayston, S.J.: Microbial functional genes involved in nitrogen fixation, nitrification and denitrification in forest ecosystems, Soil Biol. Biochem., 75, 11-25, doi:10.1016/j.soilbio.2014.03.021, 2014. Petersen, D.G., Blazewicz, S.J., Firestone, M., Herman, D.J., Turetsky, M., Waldrop, M.: Abundance of microbial genes associated with nitrogen cycling as indices of biogeochemical process rates across a vegetation gradient in Alaska, Environmental Microbiology, 14, 993e1008, 2012. van der Heijden, M.G.A., Bardgett, R.D., and van Straalen, N.M.: The unseen majority: soil microbes as drivers of plant diversity and productivity in terrestrial ecosystems, Ecol. Lett., 11, 296-310, doi:10.1111/j.1461- 0248.2007.01139.x, 2008. Wertz, S., Leigh, A.K.K., Grayston, S.J.: Effects of long-term fertilization of forest soils on potential nitrification and on the abundance and community structure of ammonia oxidizers and nitrite oxidizers. FEMS Microbiology Ecology 79, 142-154. DOI: 10.1111/j.1574-6941.2011.01204.x, 2012.

Comment #5: Line 137-139: add the data or reference Response: A related reference has been added (line 159).

Comment #6: Line 174-175: why not add rainwater? Response: Yes, it is better to add rainwater. But in our precipitation manipulation experiment, it is practically and logistically difficult to do so mainly for two reasons. First, it requires a big tank to store the rainwater excluded during the dry season in order to add them back into the plots during the following wet season; even if we have such a facility the water quality might change tremendously after 6 months of storage. Second, if we collect rainwater during the wet season and add them into plots instantaneously, it would require a forest area several times larger than the experimental plot ($\sim$150 m2) and the corresponding exclusion facilities to collect the rainwater, which are costly and labor-intensive. We

therefore used the water from a nearby pond. The chemistry of the throughfall and pond water were sampled and analyzed several times during the first year of the experiment. Results showed that total organic carbon and inorganic N contents were higher in the rainwater than in the pond water, which assures that we did not add nutrients while adding water. We added more descriptions to address this concern during this revision (see lines 205-208).

Comment #7: Line 183: in brackets, is that the instrument model of meteorological station? Response: Yes, that is the model of the rain gauge, an instrument used to measure rainfall.

Comment #8: Line 255-292: too many sentences for the introduction of Nitrous oxide (N2O) fluxes measurement Response: We eliminated several sentences that were very detailed and less important (see lines 287-293, 296-297, 300-303, 305-306, and 309-311). The remaining key descriptions are sufficient to understand the measurement method we used.

Comment #9: Line 297-298: why not use paired sample T test? Response: Thanks for the reviewer's suggestion. The paired sample T test is also called dependent T test, and the objective of this statistical method is to compare values from the same individuals at different conditions or different times. The independent sample T test aims at comparing the values from two different sets of individuals at different experimental conditions (Field, 2009). In our study, we focused on the comparison of one variable values from two different plots (i.e., precipitation change plot and control plot) at the same time. Although the soil properties and vegetation features have no significant differences between the two plots before our experiment, they are still separated from each other, and the soil environmental heterogeneity still exist between the two plots. Therefore, the compared plots cannot be assumed as identical as one plot, and the independent sample T test should be more appropriate in our situation to compare the values between treatment plot and control plot. Field, A.: Discovering Statistics Using SPSS, SAGE Publications Ltd (the third edition), London, 2009.

Comment #10: Line 395-402: Need further explanation why the precipitation addition treatment decreased soil water contents. How this SWC change will affect functional gene results. Response: The most possible reason is that the trees in the precipitation treatment plots (height: 10.2±5.0 m, DBH: 10.7±6.3 cm) are bigger than those in the control plot (height: 7.7±3.5 m, DBH: 9.5±5.2 cm), which might have caused greater soil water loss via tree transpiration in the precipitation manipulation plots during the wet season. Another reason might be the large size of precipitation events added (55 mm/day).. Large-sized precipitation is similar to flood-irrigation that can break the soil pore and lead to soil structural decline (Barber et al., 2001; Murray and Grant, 2007) and consequently affect the water retention capacity (Loll and Moldrup, 2000). More discussions have been added for the explanation of this result (lines 445-458). From our results, we did not find any direct effects of SWC or precipitation events on functional gene abundance. Instead, changes in SWC and precipitation events showed direct effects on $NO_3^-$ and $NH_4^+$ concentrations, which might indirectly influence functional genes abundance. For instance, large precipitation events could cause large soil $NO_3^-$ loss by leaching, and lower SWC could result in decreased soil $NH_4^+$ by higher ammonium volatilization (Mcgarry et al., 1987). Consequently, decreased $NO_3^-$ and $NH_4^+$ availabilities could restrict the growth of functional microorganisms and therefore result in decreases of functional microbial abundance (see Fig. 2 and 4). Barber, S.A., Katupitiya, A., and Hickey, M.: Effects of long-term subsurface drip irrigation on soil structure, Proceedings of the 10th Australian Agronomy Conference, Hobart 2001. Loll, P., and Moldrup, P.: Soil characterization and polluted soil assessment, Aalborg University, 2000. Mcgarry, S.J., O'Toole, P., and Morgan, M.A.: Effects of soil temperature and moisture content on ammonia volatilization from urea-treated pasture and tillage soils. Ir. J. agric. Res., 26(2/3):173-182, 1987. Murray, R.S., and Grant, C.D.: The impact of irrigation on soil structure, The national program for sustainable irrigation (Land & Water Australia), Braddon, 2007.

Comment #11: Fig. 1: precipitation change Fig. 2 mark the meaning of the blue bar; why the SWC is significantly lower under the precipitation addition treatment than the

control in wet season? Response: We have changed "precip-change" to "precipitation change" in Fig. 1 as suggested by the reviewer. In Fig. 2, the blue bar represents "precipitation not excluded". We have added the symbol description in the panel. The lower SWC in the precipitation addition plots was most likely due to the greater soil water loss caused by greater transpiration from relatively larger trees in the precipitation addition plots. Another possible reason was that we might had missed capturing the SWC increase after water addition since our soil sampling was at least 1 week later from the water addition date. More explanations on this concern are also provided in the Response to Reviewers Comment #10 and in the revised discussions (lines 445-458).

Comment #12: Fig. 4 why not use log transformed number? Response: Actually, we also drew the figure with log transformed data of functional microorganisms during the data processing. However, the changes in gene abundance with sampling time and the differences of gene abundance between the precipitation change treatment and control were not so obvious in visual sense as the figure displayed with original data. In order to display our data more straightforward, the figure drawn by original data was used in the current version.

Comment #13: Fig. 5 add symbols a,b,c in the figure and M.S. Response: Thanks for the recommendation. Symbols have been added to all the figures to indicate different panels. Specific figure panel symbols (e.g. Fig. 2a) have been cited in the main text as well.

Comment #14: Fig. S2 why analyze the relationships between nirK+nirS and archaeal amoA; between nosZ and nirk+nirS? Response: The objective of the correlation analyses in Fig. S2 was to detect the significant relationships between each two variables before the structural equation modeling (SEM) analyses. Based on these primary correlation results, the structural equation models could be constructed, thereafter, the casual relationships among these variables were further detected and testified. We analyzed the relationships between nirK+nirS and archaeal amoA gene

abundance; between nosZ and nirK+nirS gene abundance in Fig. S2, and found significant positive correlation between nosZ and nirK+nirS gene abundance. Thus, the relationship between nosZ and nirK+nirS was added in the SEM analyses (see Fig. 5c), and the casual relationship between the two variables was further examined. A brief description of Fig. S2 have been added in the statistical analysis part to clarify this confusion (lines 347-349).

Please also note the supplement to this comment:
http://www.biogeosciences-discuss.net/bg-2017-3/bg-2017-3-AC1-supplement.pdf

―――――――――――――――

---

## Author Comment (AC2) · 20 Mar 2017

General Comment:

This manuscript uses an interesting design to explore the complex relationships between changes in precipitation regimens and soil N dynamics. The authors monitored changes in microbial functional genes, N-mineralization rates, and soil N2O flux to understand how changes in precipitation regime, but not total precipitation amount, might affect soil biogeochemical cycles (i.e. N-cycle). Results indicated that 20% of total N-mineralization and nitrification rate were explained by microbial abundance (determined via fumigation) and soil moisture content, and that amoA abundance also helped to explain a significant portion of variation in measured N pools. Together, the results indicate that predicted changes in seasonal precipitation associated with changing climate may result in increased NO3- leaching form soils and decreases in N2O emissions. Although the manuscript is scientifically sound and interesting, it should be heavily edited for proper grammar, spelling, and sentence structure. Response: Thanks to the reviewer for considering our work interesting. We carefully polished the English of the revised manuscript (e.g., lines 79-81, 102-118, 435-438, 537-540, and 574-582). Hopefully all language issues have been fixed. We acknowledged the reviewers for their constructive comments in the Acknowledgement section. The revised manuscript is attached as a Supplement to the point-by-point response to reviewers comments. All the changes have been marked in the revised version.

Specific Comments:

Comment #1: Line 35-37: the wording of this sentence is confusing, please clarify "main factor" (i.e. is it a statistical inference that is meant here?) Response: We thank the reviewer's recommendation. This sentence has been reconstructed in a more clearly way: "More than 20% of the total variation of net nitrification and N mineralization could be explained by microbial abundance and soil water content (SWC). Noticeably, archaeal amoA abundance showed the highest correlation coefficients ($\geq$ 0.35) with net N transformation rates, suggesting the critical role of archaeal amoA abundance in determining N transformations." (see lines 36-39).

Comment #2: Line 39: do the authors mean "both dry and wet seasons"? this sentence should be clarified. Response: Yes, we meant to say "both dry and wet seasons". The sentence has been clarified as "However, N2O emission decreased moderately in both dry and wet seasons due to changes in nosZ gene abundance, MBC, net nitrification and SWC." (see line 42).

Comment #3: Line 43: what was "determined by changes in DOC and NH4+"? please

clarify this conclusion. Response: This part of the sentence meant that functional genes abundance and MBC were mainly affected by changes in soil dissolved organic carbon (DOC) and $NH_4+$ based on the results of structure equation modeling (SEM) analyses. As displayed in Fig. 5, DOC and $NH_4+$ showed significantly direct effects on all the functional genes abundance and MBC content. As much as 15 - 64% of the variation in these genes abundance and MBC content could be explained by DOC and $NH_4+$. Thus, DOC and $NH_4+$ were the main factors in affecting changes in functional genes abundance and MBC in our study. The conclusion sentence has been revised for further clarity (see lines 45-47).

Comment #4: Line 49: this sentence is missing a word, are the authors indicating that precipitation changes have been observed as increasingly sever, or that they are predicted to be increasingly severe. Response: It's been predicted to be increasingly severe over this century. The sentence has been rewritten as "Precipitation changes caused by global climate change are predicted to be increasingly severe over the century" in the revised version (see line 52).

Comment #5: Line 54: how have changes in precipitation events have exceeded other climate change-related fluctuations? Response: We meant that precipitation changes are more spatially and temporally heterogeneous compared to the other climate change factors such as temperature and atmospheric $CO_2$ concentration. Therefore the complexity and unpredictability of future precipitation changes might exceed other climate changes (Beier et al., 2012). Furthermore, the direction and types of the changes in precipitation patterns are different among ecosystems. For instance, the annual precipitation amount will decrease in subtropical ecosystems, while it will increase in the mid- and high latitudes (Mc-Carthy, 2011).This will lead to more spatially heterogeneous and less predictable of precipitation change on ecosystems than for other major climate change drivers. A meta-analysis of experimental precipitation manipulation has revealed that precipitation change effects on ecosystems are less consistent than the effects of other climate changes, such as elevated $CO_2$ and warming (Wu et al., 2011). This may reflect the more fluctuation and uncertainty of precipitation changes. Beier, C., Beierkuhnlein, C., Wohlgemuth, T., Penuelas, J., Emmett, B., Korner, C., de Boeck, H.J., Christensen, J.H., Leuzinger, S., Janssens, I.A., and Hansen, K.: Precipitation manipulation experiments - challenges and recommendations for the future, Ecol. Lett., 15, 899-911, doi:10.1111/j.1461-0248.2012.01793.x, 2012. Wu, Z.T., Dijkstra, P., Koch, G.W., Penuelas, J., and Hungate, B.A.: Responses of terrestrial ecosystems to temperature and precipitation change: a meta-analysis of experimental manipulation, Glob. Change Biol., 17, 927-942, doi:10.1111/j.1365-2486.2010.02302.x, 2011. Mc-Carthy, J.J.: Climate change 2001: impacts, adaptation, and vulnerability: contribution of Working Group II to the third assessment report of the Intergovernmental Panel on Climate Change. Cambridge University Press, 2001.

Comment #6: Line 57: seasonal redistribution of what metric? Response: We meant to say "redistribution of seasonal precipitation" (e.g., reduced dry-season precipitation but increased wet-season precipitation). The whole sentence has been revised for clarity (see line 62).

Comment #7: Line 123: reference for the SEM? Also, please introduce SEM and how/ if they have been previously used in a similar manner. Response: Two relevant references for the SEM analyses have been added (see line 141). Meanwhile, an introduction of SEM including previous examples of using this method have been added in the statistical analysis part (see lines 332-337).

Comment #8: Line 130: please clarify "easily" Response: It was not an accurate word. We have revised it as "strongly" to clarify the sentence without changing the meaning to it (see line 151).

Comment #9: Line 135-145: please add a reference for these data, or indicate how the authors collected the data. Response: A reference containing this information has been cited (see line 159). A brief description of the vegetation investigation method is also provided (see lines 162-164).

Comment #10: Line 174: was this water also examined for microbial community structure and functional genes? Is it possible that this water acted as a source of microbial change? Response: It is a very intriguing question! No, we did not examine the microbial community structure and functional genes in the water. But we suspect that the microbial community composition and functional gene abundance should be different in the rainwater and the pond water. It is difficult to speculate that whether such difference can cause soil N transformation and soil microbial community changes. Our experiment is still running, we will take the reviewer's suggestion seriously and examine whether the microbial community composition differ between the two sources of water in future studies.

Comment #11: Line 231: why did the authors not also quantify bacterial and archaeal 16S rRNA genes? The assay would independently confirm results indicated by chloroform fumigation techniques, and may potentially help to shed light on the negative results associated with amoB gene quantification. Response: The 16S rRNA and ITS genes have been quantified and reported in Zhao et al. (2017). This work was mainly focused on the roles of the key functional microorganisms in driving soil N transformations under precipitation changes. As suggested by previous studies (van der Heijden et al., 2008), ammonia-oxidizing archaea (AOA) and bacteria (AOB) marked by the amoA functional gene drive the central and rate-limiting step in N cycle: ammonium oxidation. The denitrification process is mainly driven by nitrite-reducing bacteria marked by the nirK and nirS genes and nitrous oxide-reducing bacteria marked by nosZ gene (Chan et al., 1997; Billings, 2008). Thus, we only quantified these functional microorganisms in our study. The negative results of AOB community abundance were mainly caused by the low pH value (4.08 $\pm$ 0.05) in our study soil, which has been evidenced previously (Isobe et al., 2012) and described in the manuscript (see lines 394-397). Except for the functional microorganisms, the chloroform fumigation extracted microbial biomass carbon (MBC) was also investigated and evidenced as an important factor in driving N transformation in our study. There are two seasons: firstly, the chloroform fumigation techniques can measure the total microbial biomass including bacteria, archaea and fungi (Wu et al., 1990). Therefore, the MBC determined by the chloroform fumigation method is more appropriate than the specific gene copy numbers to reflect the changes in overall microbial community size in soil. Secondly, since soil microbial biomass can be either a source or sink of available nutrients, it usually plays an important role in soil nutrient transformations (Singh et al., 1989). Chan, Y.K., McCormick, W.A., Watson, R.J: A new nos gene downstream from nosDFY is essential for dissimilatory reduction of nitrous oxide by rhizobium (sinorhizobium) meliloti. Microbiology 143, 2817-2824, doi:10.1099/00221287-143-8-2817, 1997. Billings, S.A.: Biogeochemistry: nitrous oxide in flux. Nature 456, 888-889, 2008. Isobe, K., Koba, K., Suwa, Y., Ikutani, J., Fang, Y.T., Yoh, M., Mo, J.M., Otsuka, S., and Senoo, K.: High abundance of ammonia-oxidizing archaea in acidified subtropical forest soils in southern China after long-term N deposition, Fems Microbiol. Ecol., 80, 193-203, doi:10.1111/j.1574-6941.2011.01294.x, 2012. Singh, J.S., Raghubanshi, A.S., Singh, R.S., and Srivastava, S.C.: Microbial biomass acts as a source of plant nutrients in dry tropical forest and savanna, Nature, 338, 499-500, doi:10.1038/338499a0, 1989. van der Heijden, M.G.A., Bardgett, R.D., and van Straalen, N.M.: The unseen majority: soil microbes as drivers of plant diversity and productivity in terrestrial ecosystems, Ecol. Lett., 11, 296-310, doi:10.1111/j.1461- 0248.2007.01139.x, 2008. Wu, J., Joergensen, R.G., Pommerening, B., Chaussod, R., Brooks, P.C.: Measurement of Soil Microbial Biomass C by Fumigation- Extraction - An Automated Procedure, Soil Biol. Biochem., 22, 1167-1169, 1990. Zhao, Q., Jian, S., Nunan, N., Maestre, F. T., Tedersoo, L., He, J., Wei, H., Tan, X., and Shen, W.: Altered precipitation seasonality impacts the dominant fungal but rare bcterial taxa in subtropical forest soils. Biol. Fertil. Soils, 53, 231-245, doi: 10.1007/s00374-016-1171-z, 2017.

Comment #12: Line 381: could the opposite also be true here? That alterations in DOC and N are due to changes in functional gene abundance? Response: Yes, we agree with the reviewer that soil DOC and N could be affected by changes in functional gene abundance. In our study, the changes in NH4+ and NO3- availabilities were mainly affected by soil water content (SWC) and precipitation events, while the DOC

changes could not be explained by the measured variables (see Fig. 5). However, the alterations in functional gene abundance were largely explained by soil DOC and N changes (see Fig. 5). Thus, the effects of functional gene abundance on soil DOC and N were not emphasized in the discussion part.

Comment #13: Line 398: reference? Response: The reference (Gao et al., 2017) has been added (see line 453).

Comment #14: Line 424: is there a figure or table that would also demonstrate this statement? Response: Yes, this statement could be demonstrated by the changes in SWC (Fig. 2d), MBC (Fig. 2h), N transformation rates (Fig. 3b, d) and archaeal amoA gene abundance (Fig. 4b), and the relationships between net N transformation rates and these variables (i.e., SWC, MBC and amoA gene abundance) in Fig. S2. We have added the Figure numbers behind the corresponding text (see lines 485-488).

Comment #15: Line 433: please clarify this sentence, how are microorganism responsible for reducing microbial abundance? Response: This sentence has been clarified (see lines 495-501). When the microorganisms are limited by substrate resources, such as available C and N, one part of microorganisms may die from starvation or competition. Since microbial biomass C and N can be either a sink or source of available nutrients (Yang et al., 2010), the other part of microorganisms which survived from starvation and competition can reuse the C and N released from the dead microorganisms (Borken and Matzner, 2009). Finally, the community size of soil microorganisms may reduce by the death of some microorganisms after substrate decrease. Borken, W., and Matzner, E.: Reappraisal of drying and wetting effects on C and N mineralization and fluxes in soils, Glob. Change Biol., 15, 808-824, doi:10.1111/j.1365-2486.2008.01681.x, 2009. Yang, K., Zhu, J.J., Zhang, M., Yan, Q.L., and Sun, J.X.: Soil microbial biomass carbon and nitrogen in forest ecosystems of Northeast China: a comparison between natural secondary forest and larch plantation, Journal of Plant Ecology, 3, 175-182, doi:10.1093/jpe/rtq022, 2010.

Comment #16: Line 444: are there other appropriate references available here? Response: Yes, we added one more reference here (Gao et al., 2016, see line 513) as suggested by the reviewer.

Comment #17: Figures 2-5 would benefit from the addition of panel indicators (i.e. "Fig1 A") to help clarify to the reader exactly which data the authors are discussing in the MS. Fig 5: this is a nicely illustrated complex figure. Response: We thank the reviewer's positive comment on Fig. 5. The panel indicators have been added for all the figures throughout the manuscript.

Please also note the supplement to this comment:
http://www.biogeosciences-discuss.net/bg-2017-3/bg-2017-3-AC2-supplement.pdf

---

## Author Comment (AC3) · 20 Mar 2017

General Comment:

The manuscript bg-2017-3 reported soil N transformation rates in relate to soil properties and microbial functional gene abundance in a rainfall-manipulation experiment in a subtropical forest. The authors showed that the rainfall manipulation (dry-season reduction and wet-season addition) increased $NO_3^-$ leaching and had minor effect on $N_2O$ emission, which can be linked to changes in soil properties and specific microbial functional genes (by SEM). This topic is very relevant to Biogeosciences, and the results are robust based on solid data (monitoring of soil properties, in situ N transformation rate measurements, and microbial functional gene measurements). I have few suggestions for the authors to consider in revision. Response: We thank the referee for his/her valuable comments. All the questions and suggestions provided by the reviewer are really helpful for the improvements of our manuscript. We acknowledged the reviewers for their constructive comments in the Acknowledgement section. The revised manuscript is attached as a Supplement to the point-by-point response to reviewers comments. All the changes have been marked in the revised version.

Specific Comments:

Comment #1: To be more accurate, DOC is EOC (extractable organic carbon). Response: Agreed. DOC has been changed to EOC throughout the manuscript.

Comment #2: The writing should be improved for clarity and readability. In some sections, it is wordy and not easy to follow the logic. Response: The writing has been carefully revised in the new version. The long sentences have been rewritten with short and clear sentences. Some sections (e.g., descriptions on N2O measurement methods have been largely shortened (see lines 286-311).

Comment #3: Microbial functional gene abundance has limited power in explaining the N transformation rates: a) many genes are involved in a process, and b) a gene is there does not mean it is expressed or "functioning". This needs to be mentioned. Response: Many thanks for the constructive comment. These points have been mentioned in the section where the contributions of functional gene abundance to N transformation rates are discussed (see lines 522-529)

Comment #4: L397-402: Why select these 8 plots for the experiment? The initial differences in stand characteristics between treatments should be minimized for such experiment. Response: These 8 plots was assigned randomly to minimize the spatial variation of soil properties. Actually, we compared the stand characteristics (i.e., species composition, tree height, tree number, DBH and crown width) between the precipitation manipulation plots and the control plots prior to the experiment, and found no significant differences in these characteristics. The information of stand characteristics of the two plots have been described in the manuscript (see lines 174-180).

Comment #5: Figure 1 and 5: why soil pH was not included in the SEM? Was it measured? Response: We only measured the soil pH before and after our experiments, and there were no changes in soil pH caused by the treatments either in the dry or wet season. Thus, we assumed that soil pH was not an import factor in driving the changes of functional gene abundance and N transformation rates in our treatments. Therefore, soil pH was not involved in the SEM analysis. The general information of soil pH has been added in the results section (see lines 378-383).

Comment #6: Figure 2: Why rainfall addition in the wet season did not lead to expected increase in soil water content, in both years? Also, MBC was not significantly affected by the rainfall treatment even in the dry season, which is unexpected. Do you have continuous measurement of soil moisture (by TDR or Decagon sensors) in these 8 plots? Response: No, we did not monitor soil moisture continuously during the experiment. That is probably one reason why adding water actually did not rise soil moisture: our manual sampling of soils usually lagged 1 week behind the date of water addition; we therefore missed capturing the moisture changes caused by the water addition. There may be other reasons for the result of lower soil water content in water addition plots. The larger trees in the precipitation addition treatment plots (height: 10.2±5.0 m, DBH: 10.7±6.3 cm) may have greater transpiration rate than the trees in control plot (height: 7.7±3.5 m, DBH: 9.5±5.2 cm) in summer, which might have caused greater soil water loss in the water addition plots compared to the control plots. We have tried to minimize the stand variation by selecting the plots with similar vegetation features before the experiment, but it was difficult to find 8 plots with the same stand characteristics in field. Secondly, more than 55 mm water was added each time in the wet season which might result in flood-irrigation in the precipitation manipulation plots. As suggested by previous studies, flood-irrigation could break the
soil pores and lead to soil structural decline (Barber et al., 2001; Murray and Grant, 2007), which may affect the water retention capacity, as soil water retention capacity is relate to pore-size and pore-distribution (Loll and Moldrup, 2000). More discussions have been added to address such concerns (see lines 445-458). We expected that MBC would decrease under the precipitation reduction treatment. But actually MBC was not significantly changed by the 67% of precipitation reduction during the dry season. This result is also confirmed by the unaffected total microbial phospholipid acids (PLFAs) as reported in Zhao et al. (2017). We argue that the main reason was that soil moisture reduction (10-21%) was not as severe as we expected – it only decreased by 10-21% in responding to a 67% precipitation exclusion. Such a moderate reduction of soil moisture might have not reached the point at which microbial growth and other activities can be limited. Although the total microbial biomass was not changed, but the composition of the microbial community was altered, which have been reported in Zhao et al. (2017). Barber, S.A., Katupitiya, A., and Hickey, M.: Effects of long-term subsurface drip irrigation on soil structure, Proceedings of the 10th Australian Agronomy Conference, Hobart 2001. Loll, P., and Moldrup, P.: Soil characterization and polluted soil assessment, Aalborg University, 2000. Murray, R.S., and Grant, C.D.: The impact of irrigation on soil structure, The national program for sustainable irrigation (Land & Water Australia), Braddon, 2007. Zhao, Q., Jian, S., Nunan, N., Maestre, F. T., Tedersoo, L., He, J., Wei, H., Tan, X., and Shen, W.: Altered precipitation seasonality impacts the dominant fungal but rare bcterial taxa in subtropical forest soils. Biol. Fertil. Soils, 53, 231-245, doi: 10.1007/s00374-016-1171-z, 2017.

Comment #7: Table S2: stand characteristics (species composition, stem density, tree height, basal area, etc.) should be included. Response: The information of vegetation characteristics for the precipitation manipulated plots and control plots has been added in the manuscript (see lines 174-180).

Please also note the supplement to this comment:
http://www.biogeosciences-discuss.net/bg-2017-3/bg-2017-3-AC3-supplement.pdf

[Figure]

**Supplement:**

[revised manuscript text omitted]

---

## Author Response (AR1)

**Response to Associate Editor's comments**

**Manuscript #: bg-2017-3**

**Associate Editor: Dr. Denise M. Akob**

**Comment #1:** Thank you for submitting your Research Article titled "Soil nitrogen transformation responses to seasonal precipitation changes are regulated by changes in functional microbial abundance in a subtropical forest" to Biogeosciences. The 3 referees provided thoughtful, constructive comments on your paper, and in your responses and revisions you have adequately addressed all major and minor issues. However, I have a few minor revisions (listed below) that should be addressed before acceptance. Please upload a revised version with these and the reviewers' comments addressed.

**Response:** Thank you very much for considering our revisions adequate and for providing valuable suggestions to further improve our manuscript. We have carefully studied each of your comments and incorporated them into this revision. The point-by-point responses to your comments are listed below and marked in the revised manuscript. We hope that you would find these revisions satisfactory.

**Comment #2:**
    L. 219: write out "PC"
    L218- elsewhere: write out the full month name, don't abbreviate.
**Response:** Done as suggested. Please see lines 198, 199, 202.

**Comment #3:** L. 259: do you mean microliters and not milliliters for the qPCR volume?
**Response:** Thanks for detecting this. It should be microliters, the unit has been corrected. Please see line 238.

**Comment #4:** L. 262: was this PCR- or molecular water?
**Response:** It was RNase free Ultra-Pure water, which could be used in PCR protocol. The term "double-distilled" has been revised. Please see line 241.

**Comment #5:** L. 265: please provide a reference for the standard construction and/or provide more details. E.g., what sequences were used in the plasmids?
**Response:** The standards were constructed using the method described in Henry et al. (2006) and Isobe et al. (2011). Briefly, to generate the standard curves, the target gene fragment of the soil clone obtained in this study were used. For example, we obtained the archaeal *amoA* PCR product with the same primers used in real-time PCR (i.e., CrenamoA 23f/CrenamoA 616r) and the extracted soil DNA as template. The archaeal *amoA* PCR product was cloned into the pMD20-T vector (TaKaRa, Dalian Division), and the cloning fragments were transformed into *Escherichia coli* JM109 strains. The recombinant *Escherichia coli* JM109 strains carrying the archaeal *amoA* recombinant plasmids were inoculated into LB broth with ampicillin and incubated at 37°C overnight. The plasmid DNA was then extracted using the Hipure Plasmid Mini Kit (Magen, Guangzhou, China) and quantified on a NanoDrop 2000 spectrophotometer (Thermo Fisher Scientific Inc., USA). The presence of archaeal *amoA* inserts was verified by PCR with the same primers (i.e., CrenamoA 23f/CrenamoA 616r) and gel electrophoresis. The copy numbers of the standard archaeal *amoA* gene numbers was expressed as the DNA copy numbers of the extracted plasmid DNA carrying archaeal *amoA* gene, which was calculated from the plasmid DNA size, concentration, and average base pair molecular weight. Standard curve was then generated from a tenfold serial dilution ($10^3$-$10^8$ copies per µl) of the plasmid DNA. The references and more sentences have been added to describe the standard construction as suggested. Please see lines 242-255.

[revised manuscript text omitted]